# Monopoly of Minor Group Rhinovirus Infections in Hospitalised Children in Hong Kong During the SARS-CoV-2 Pandemic

**DOI:** 10.3390/v17101316

**Published:** 2025-09-28

**Authors:** Jason Chun Sang Pun, Kin Pong Tao, Shaojun Liu, Ben Kam San Wong, Tony Chun Hei Lei, Lucky Lu Yi Tsoi, Joseph Gar Shun Tsun, Agnes Sze Yin Leung, Paul Kay Sheung Chan, Renee Wan Yi Chan

**Affiliations:** 1Department of Paediatrics, Faculty of Medicine, The Chinese University of Hong Kong, 6/F Lui Che Woo Clinical Sciences Building, Prince of Wales Hospital, Shatin, New Territories, Hong Kong, China; jason@link.cuhk.edu.hk (J.C.S.P.);; 2Hong Kong Hub of Obstetric and Paediatric Excellence, The Chinese University of Hong Kong, 8/F Research Office, Tower A, Hong Kong Children’s Hospital, 1 Shing Cheong Road, Kowloon Bay, Kowloon, Hong Kong, China; 3Laboratory for Paediatric Respiratory Research, Li Ka Shing Institute of Health Sciences, Rm 807, Li Ka Shing Medical Sciences Building, The Chinese University of Hong Kong, Prince of Wales Hospital, Shatin, New Territories, Hong Kong, China; 4SH Ho Research Centre for Infectious Diseases Faculty of Medicine, Rm 207, 2/F, S.H. Ho Research Centre for Infectious Diseases, JC School of Public Health Building, Prince of Wales Hospital, Shatin, New Territories, Hong Kong, China; 5Department of Microbiology, Faculty of Medicine, The Chinese University of Hong Kong, 1/F Lui Che Woo Clinical Sciences Building, Prince of Wales Hospital, Shatin, New Territories, Hong Kong, China

**Keywords:** rhinovirus, RV-A49, transmissibility, environmental stability, air–liquid interface, Hong Kong

## Abstract

Background: While rhinoviruses (RVs) typically cause mild respiratory infections, their persistence during the SARS-CoV-2 pandemic, particularly in Hong Kong’s strict zero-coronavirus disease 2019 policy, revealed unexpected epidemiological patterns. Two distinct RV surges emerged despite stringent public health measures, suggesting unique transmission advantages among circulating strains. We hypothesised that RV persistence during pandemic restrictions reflected strain-specific adaptations in respiratory tract replication efficiency and/or immune evasion. Methods: We analysed RV genotypes and conducted blinded clinical severity assessment for 96 paediatric hospitalisations during 2020–2021 outbreaks, compared with 180 age- and sex-matched control subjects from the corresponding weeks in pre-pandemic years (2018–2019). RV isolates from 2020 to 2021 outbreaks were characterised for their replication competence and transcriptomic responses in primary human nasal epithelial cell (HNEC) and environmental stability assays, using RV-A16 and RV-A1B as controls. Result: Minor group genotypes RV-A47 and RV-A49 were overrepresented during these two outbreaks. RV-A49 exhibited comparable replication efficiency to RV-A16 but induced significantly stronger transcriptomic responses, notably enhanced TNF and IL-1 signalling, in HNECs, alongside robust replication competence. Our data also suggests the association of RV-A49 with tachypnoea in 2021, particularly in younger males, though limited by a small sample size and single-centre design. Conclusion: The predominance of RV-A49 in hospitalised children during the SARS-CoV-2 pandemic potentially driven by its replication competence in HNECs and its capacity to enhanced inflammatory responses. The result is hypothesis-generating, warranting further studies with historical strains and broader populations to confirm strain-specific severity.

## 1. Introduction

The SARS-CoV-2 pandemic ushered in an era of unprecedented public health interventions globally, with Hong Kong imposing particularly rigorous measures from 2020 to 2022. These included stringent border controls, mandatory quarantine and contact tracing, widespread compulsory mask-wearing, and a heightened public awareness of hand hygiene. While these strategies have been shown to significantly reduce SARS-CoV-2 transmission and COVID-19 mortality across diverse populations and settings [1,2], they also had a profound impact on the epidemiology of other respiratory viruses [3]. Many common respiratory pathogens, such as influenza and parainfluenza viruses, virtually vanished from routine surveillance programmes worldwide during this period. However, amidst this landscape of viral suppression, rhinovirus (RV) remained prevalent in the community. Moreover, a surge of rhinovirus was identified during the reopening of cities and schools [4,5,6], which led us to a research question on the unique niche of RV in overcoming the public health measures and perhaps an early re-emergence, which was not seen in other respiratory viruses.

RVs are among the most common viral pathogens affecting humans and are the leading cause of the common cold. Beyond mild upper respiratory tract infections, repeated wheezing illnesses caused by RV in early life are the strongest predictor of asthma development by age 6 [7] and by age 13 [8], and increasingly recognised for their role in exacerbating chronic respiratory diseases, such as asthma and chronic obstructive pulmonary disease, and in triggering lower respiratory tract infections, particularly in vulnerable populations including children, the elderly, and immunocompromised individuals. RVs are classified into three species—RV-A, RV-B, and RV-C—based on genetic and antigenic characteristics. Within RV-A and RV-B species, RVs are further grouped by their receptor usage, where major group RVs utilise intercellular adhesion molecule-1 (ICAM-1) for cell entry and minor group RVs use members of the low-density lipoprotein receptor (LDLR) family. RV-C, which is genetically distinct, uses cadherin-related family member 3 (CDHR3) as its receptor.

Hong Kong experienced two distinct surges of respiratory infections predominantly associated with RV. The first surge of upper respiratory tract infections occurred from October to November 2020, immediately after the reopening of nursery schools, childcare centres, kindergartens, and primary and secondary schools. This outbreak affected over 2000 students and staff [9], leading to territory-wide school closures for all kindergartens and childcare centres in November. During this period, the percentage of RV-positive cases surged. A second soar in RV infections was observed from April to August 2021. This indicated that the typical seasonal spring and autumn peaks of EV activity remained unhindered, even during the exceptional lockdown period with enhanced personal hygiene practices [10].

This study was designed to investigate the underlying factors contributing to these specific episodes of RV outbreak in children during the SARS-CoV-2 pandemic. Specifically, we aimed to explore if a particular genotype was responsible for these surges. To evaluate the replication competence of the circulating RV strains in comparison to the conventional RV strains, virus isolation and experimental infection studies were performed using well-differentiated primary human nasal epithelial cells (HNECs). Furthermore, to address the unrestricted transmissibility of RV during this critical period, we investigated the environmental stability and transcriptomic changes induced in HNECs by the isolated RV strains.

## 2. Materials and Methods

Clinical sample selection criteria and viral RNA extraction and genotyping were as follows. RV genotyping of respiratory samples: Frozen archives of nasopharyngeal swab (NPS) from hospitalised children aged 0 to 17 years, admitted to Prince of Wales Hospital between week 44 to 48 in 2020 (autumn cohort) and week 16 to 32 in 2021 (spring cohort), were retrieved and genotyped. Nasopharyngeal aspirates (NPAs) from the same period in 2018 and 2019, where enterovirus/rhinovirus (EV/RV) was the sole detectable pathogen, were selected as age- and sex-matched controls in a 1:1 case-to-control ratio. To reduce the potential effect of having co-infection cases, multiplex PCR assay was performed to check for the presence of human metapneumovirus, adenovirus, influenza A and B viruses, parainfluenza viruses (PIV-1, -2, -3, -4), respiratory syncytial virus, and SARS-CoV-2 (for samples post-2020). Only samples testing positive for EV/RV and negative for all other pathogens were included, ruling out co-infections with these viruses. Viral RNA was extracted using MiniBEST Viral RNA/DNA Extraction Kit, followed by reverse transcription with PrimeScript RT Master Mix (TaKaRa, Kusatsu, Japan). The cDNA was amplified via nested PCR targeting the VP4 and VP2 regions using outer primers F2 [5′-CCG GCC CCT GAA TGY GGC TAA-3′] and -R2 [5′-ACA TRT TYT SNC CAA ANA YDC CCA T-3′], with the cycle condition as follows: 40 cycles of 98 °C for 10 s, 55 °C for 30 s, 72 °C for 45 s, followed by inner primers F3 [5′-ACC RAC TAC TTT GGG TGT CCG TG-3′] and R3 [5′-TCW GGH ARY TTC CAM CAC CAN CC-3′], with cycling condition of 30 cycles of 98 °C for 10 s, 55 °C for 30 s, 72 °C for 45 s. Amplicons were subjected to Sanger sequencing and compared to prototype sequences in GenBank using BLAST+ 2.7.1 for genotype identification.

Cell culture and virus isolation: H1-HeLa cells (ATCC CCL-2) were cultured in minimum essential media with 10% foetal bovine serum and 1% penicillin and streptomycin. Laboratory RV strains RV-A16 and RV-A1B were obtained from ATCC (Manassas, VA, USA). A total volume of 500 μL of NPS or NPA was filtered using 0.22 μm Spin-X columns (Corning, NY, USA) and inoculated onto H1-HeLa for 2 h at 34 °C. The inoculum was removed, cells were washed with phosphate-buffered saline (PBS), and fresh medium was added. Cultures were incubated at 34 °C in a 5% CO_2_ incubator, with cytopathic effects (CPEs) monitored daily for up to 7 days. Supernatants were harvested and subjected to at least three serial passages. The supernatant from each passage was analysed by qPCR to detect increases in viral gene copy number as evidence of successful propagation, followed by a viral titration assay to confirm the presence of infectious virions. If no viral gene copies were detected after three passages, a second attempt was performed using the original clinical specimen. Isolation was deemed unsuccessful if no viral gene copies were detected in the second attempt.

Virus quantification by viral titration and quantitative PCR: 2 × 10^4^ cells/well of H1-HeLa were seeded in 96-well plates in minimum essential media with 10% foetal bovine serum and 1% penicillin and streptomycin a day prior to the viral titration assay. Cells were washed once with PBS, and virus samples were titrated in serial half-log_10_ dilutions with the culture medium. A total volume of 150 uL of diluted viruses were added to the cell plates in quadruplicate, and CPEs were monitored daily until day 7. The 50% tissue culture infectious dose (TCID_50_) per mL was calculated using the Kärber method. For viral gene copy quantification, RNA was extracted from infected air–liquid interface (ALI) culture supernatants as described below and analysed by RT-qPCR with One Step TB Green^®^PrimeScript™ RT-PCR Kit II (TaKaRa, Kusatsu, Japan) with primers targeting the 5′ UTR Forward [5′-CCTCCGGCCCCTGAAT-3′] and Reverse [5′-AAACACGGACACCCAAAGTAGT-3′]. Quantification of RV virus was based on standard curves constructed using a 10-fold serial dilution of 5′UTR plasmid.

Differentiation and infection of ALI-differentiated human nasal epithelial cells (HNEC): Primary HNECs were collected from paediatric donors by brushing the inferior nasal turbinate with flocked swabs, with the cells expanded and differentiated as previously described [11]. The differentiation state of the ALI cultures was monitored from day 0 to day 28, with cultures considered mature from day 28 onwards. Mature ALI-HNECs were washed five times with PBS and apically infected with RV at a multiplicity of infection (MOI) of 0.01, calculated using TCID_50_ units, or filtered NPA for 2 h. Infected cells were washed and replenished with fresh medium in the basolateral compartment and maintained at 34 °C or 37 °C. Supernatants were collected by incubating 200 μL of PBS in the apical compartment for 20 min at 2, 24, 48, 72, 96, 120, 144, and 264 h post infection (hpi).

Environmental stability: Tested materials (wood, glass, stainless steel, acrylic, paper, and inner/outer mask layers) were cut precisely into 2 cm × 0.5 cm coupons using a laser cutter or snapNcut (Brother, Nagoya, Japan). Disinfection of wood, glass, stainless steel, and plastic coupons were performed by rinsing with mild detergent, followed by distilled water, and oven-dried. Paper, and mask inner- and outer-layer coupons were sterilised by autoclaving. Triplicate coupons were placed in a 12-well plate and 10 μL of RV with known titre was pipetted onto the centre of the coupon, followed by incubation at 37 °C under constant humidity in the dark. The virus was recovered by soaking the coupon in 200 μL PBS for 30 min at designated time points in triplicates.

Bulk RNA-sequencing and transcriptome analysis: Total RNA from HNECs was extracted and subjected to strand-specific library preparation with polyA enrichment, followed by sequencing using Illumina Nova6000 (Novogene, Beijing, China). Raw reads were trimmed using the Quality Score method and mapped to hg38 genome (Ensembl 104) using HISAT2. Low count features were excluded with raw count < 10. Differential expression analysis was performed using DESeq2, and differentially expressed genes (DEGs) were defined by a false discovery rate (FDR) < 0.05 and a fold change (FC) > 2. Functional enrichment analysis using Gene Ontology and Kyoto Encyclopaedia of Genes and Genomes by the Gene Set Enrichment Analysis (GSEA), with FDR < 0.05 as the significance threshold.

Statistical analysis: Epidemiological analysis was performed using the 2020 and 2021 incidences in comparison to that of the 2018–2019 pre-pandemic period using Pearson’s chi-squared test to investigate the distribution of RV species (RV-A, -B, and -C) and groups (major and minor). Bonferroni correction was used to account for multiple testing. Standardised residuals were computed post hoc to identify specific over- or under-representation. These were performed using scip.ststs module in Python (v3.9). The viral titres and RT-qPCR data were expressed as mean and standard error and were compared using 2-way ANOVA followed by Tukey post test using Prism 9 (GraphPad Software, Boston MA) USA. Statistical difference was deemed significant at *p* < 0.05 unless otherwise specified.

## 3. Results

### 3.1. Monopoly of RV During SARS-CoV-2 Pandemic

This study investigated the genetic diversity of RV during two distinct RV outbreaks that occurred in Hong Kong during the autumn (44th to 48th week) of 2020 and spring to summer (16th to 32nd week) of 2021. To assess changes in RV molecular epidemiology, RV-positive respiratory specimens were collected from patients presenting with respiratory tract infections during these periods. A total of 55 and 41 RV-positive NPSs were collected in the respective periods. For comparative analysis, age- and sex-matched control subjects from the corresponding weeks in pre-pandemic years (2018–2019) were included (Table 1).

A total of 276 specimens were included in this study and RV was successfully genotyped in 224 cases (81.16%). Molecular genotyping revealed a significant shift in RV species distribution during the pandemic. RV-A was found to be significantly more prevalent in the autumn cohort in 2020 compared to the pre-pandemic years of 2018 and 2019 (*p* < 0.0001). Furthermore, minor group RVs, which utilise low-density lipoprotein (LDLR) receptor as a viral receptor, were significantly overrepresented in 2020, as determined by Fisher’s exact test.

During the autumn 2020 outbreak, RV-A47, a minor group RV, was the dominant genotype, accounting for over 79.6% of all the RV detected during this period. Similarly, in the spring 2021 outbreak, RV-A49, another minor group RV, was the dominant genotype contributing up to 51.4% of all the detected RVs. This contrasts sharply with the pre-pandemic period (2018 and 2019), which exhibited a much higher molecular diversity of RVs (Figure 1).

Despite marked shifts in RV genotype dominance, comparative analysis of clinical severity revealed no overt increase during the two RV outbreaks amid the SARS-CoV-2 pandemic relative to pre-pandemic years. This observation suggests that the dominant minor group RVs (RV-A47 and RV-A49) may exhibit enhanced stability or transmissibility traits that allowed them to monopolise RV circulation during the unique epidemiological conditions of SARS-CoV-2 pandemic in Hong Kong.

### 3.2. Clinical Outcomes of RV Inpatient Cohorts

To investigate if the observed shifts in RV genotype dominance correlated with changes in disease severity, clinical outcomes were assessed for the 2020 RV inpatient cohort and compared with previous years. Physicians (S.L. and A.S.Y.L.), blinded to genotyping results, analysed the clinical outcomes of these patients, grading them using a modified respiratory symptom scorecard (Appendix A Table A1 [12]). Most clinical parameters, such as the duration of hospitalisation and respiratory symptom scores were not significantly higher than the pre-pandemic controls. The only exception was observed in 2021, where the percentage of the severe respiratory score (score > 5) was significantly higher than that in the control years of 2018 and 2019, contributed by a marked increase in incidence presented with tachypnoea (Appendix A Table A2). Interestingly, among these five subjects with tachypnoea, four of them were infected with RV-A49 and one with RV-C8.

### 3.3. Divergent Replication Capacities of RV-A47 and RV-A49 in Cell Culture

To further evaluate the biological niches of the two dominating RV genotypes (RV-A47 and RV-A49), clinical specimens containing these genotypes were inoculated into H1-HeLa cell cultures for virus isolation, respectively. While a clinical isolate RV-A49 was successfully isolated, demonstrating CPE after 72 h of sample inoculation and yielding a Ct value of 13, RV-A47 (n = 11) containing NPSs did not yield a successful virus isolation, with no CPE observed after six passages in H1-HeLa cells.

### 3.4. RV-A49 Replicates in HNECs in Vitro at Both 34 °C and 37 °C

To assess the replication competence and potential transmissibility of RV-A49 (the predominant minor group genotype identified in our study), we conducted controlled experimental infections at an MOI of 0.01 using HNECs at both 34 °C or 37 °C, representing the temperature of the upper and lower respiratory tract, respectively. Laboratory strains of RV-A16 (major group) and RV-A1B (minor group) were included as controls to establish baseline replication profiles for well-characterised rhinoviruses. Both RV-A16 (Figure 2A) and RV-A1B (Figure 2C) exhibited robust replication in HNECs, as evidenced by increasing viral loads measured through viral titration assays and qPCR of the supernatant collected from the apical compartment (Figure 2B,D), which reflected released viruses. Notably, both viruses showed a preference for replication at 34 °C over 37 °C, with consistent results across both assays. In contrast, RV-A49 produced no detectable infectious viral load by titration assay under identical conditions (Figure 2E), despite qPCR indicating an increase in viral gene copies for RV-A49 to RV-A16 and RV-A1B. Unlike the two laboratory strains, the H1-HeLa-adapted RV-A49 displayed no temperature preference (Figure 2F, solid lines). Additionally, RV-A47, another minor group RV, could not be isolated and propagated in H1-HeLa cells, precluding an MOI-matched experiment. However, direct inoculation of HNECs with RV-A47 containing clinical specimen yielded a 5 log_10_ increase in viral gene copy number by qPCR (Appendix A Figure A1), though no infectious titre was detected. These findings indicated that both RV-A49 and RV-A47 could replicate productively in HNECs, as evidenced by viral gene measurements. The absence of infectious titres in titration assays for RV-A49 and RV-A47 may reflect impaired progeny replication in undifferentiated non-respiratory-origin H1-HeLa cells without prior adaptation, potentially limiting their detection by standard assays.

### 3.5. RV-A49 Infection Exerted Greater Transcriptomic Changes than Laboratory Strain RV-A16 in Primary HNECs

To elucidate the mechanism underlying the observed a subtle increase in the clinical severity of RV-A49 infections relative to the pre-pandemic period (Appendix A Table A2), HNECs were infected with the isolated RV-A49 at 37 °C and harvested at 48 hpi for bulk RNA sequencing. Transcriptomic were compared to those induced by the established laboratory strain RV-A16. Viral reads of samples infected with RV-A16 (0.2%, n = 5) and RV-A49 (0.15%, n = 2) constituted a small fraction of the total mappable transcripts. Principal component analysis identified a virus-induced shift in clustering (Figure 3A), and DEG analysis demonstrated that RV-A49 induced more pronounced transcriptomic changes compared to RV-A16 relative to mock-infected control (Figure 3B). While most DEGs identified in RV-A16 overlapped with those in RV-A49 (upregulated gene: 185/188; downregulated gene: 2/6 (Figure 3C), the magnitude of expression changes was greater for RV-A49 (Figure 3B), suggesting a potential basis for its increased clinical severity. GSEA revealed that both RV strains induced classical antiviral responses such as interferon and chemokine production, as well as ISG15, JAK, and TLR signalling pathways (Figure 3D). Interestingly, RV-A49 uniquely enriched pathways related to TNF and IL-1 signalling and Dectin-2 families, which are primarily associated with mucin production. Additionally, pathways linked to ribosomal/viral translation and histone demethylation were exclusively negatively enriched in RV-A49-infected HNECs, but not in RV-A16-infected HNECs, despite comparable infectivity (Figure 3E). Taken together, these findings suggest that RV-A49 exerted greater transcriptomic response than RV-A16, potentially contributing to its heightened clinical impact despite similar infection levels in HNECs.

### 3.6. Environmental Stability

To study the environmental stability of the RV under conditions mimicking real-world surfaces, thermal inactivation experiments were conducted to examine if RV-A49 possesses exceptional environmental stability compared to RV-A16 and RV-A1B during the stringent personal hygiene pandemic period.

The decay of the infectivity of the virus was quantified by fitting data to a one-phase decay model (Figure 4). The rate constant K represents the rate of decay, and the half-life time was also calculated. Among the seven materials tested, inner and outer mask layers, acrylic, paper, glass, wood, and stainless steel, RV-A16 lab strain exhibited the slowest decay rate across all surfaces, followed by RV-A1B and RV-A49. RV-A49 did not demonstrate superior environmental stability compared to the major and minor group representatives. Infectivity decayed most slowly in masks, acrylic, and paper, followed by glass and wood, with the fastest inactivation observed on stainless steel (Appendix A Table A3). No infectious viruses were recovered beyond one-day post inoculation. These results suggest that RV-A49 does not exhibit enhanced environmental persistence compared to the other genotypes.

## 4. Discussion

This study provides a unique opportunity to capture a shift in RV circulation during the SARS-CoV-2 pandemic, where strict public health measures (e.g., masking, social distancing) altered respiratory virus epidemiology. Belonging to non-enveloped virus types, RVs are resistant to alcoholic disinfectant and less impacted by face-masking [13]. Our study provides a comprehensive analysis of RV epidemiology, genotype dominance, replication competence, and clinical outcomes during two distinct RV outbreaks in Hong Kong, one in autumn 2020 and another in spring–summer 2021, compared to the age- and sex-matched controls of the pre-pandemic years (2018–2019).

RVs exhibit extensive genetic diversity with over 170 genotypes, and the limited antigenic cross-reactivity amongst them contributes to their persistent and diversified burden in the community [14]. It can also classified into major and minor groups, cause respiratory infections, and exacerbate asthma or bronchiolitis in children [15,16]. The perception of RVs as causing ‘mild’ illness reflects their position relative to higher-morbidity viruses like influenza or RSV, thereby underestimating their capacity for clinical consequences. Minor groups RV-A47 and RV-A49 caused hospitalisations, with RV-A49 preliminarily linked to tachypnoea (4/5 severe cases, aged 1.5–11 years, all male), suggesting severity in younger males, though limited by small sample size (n = 5) and male bias (vs. 50.9% male in RV-positive cohort).

During the SARS-CoV-2 pandemic, numerous non-pharmaceutical interventions were imposed to curb the spread of not only SARS-CoV-2 but also other respiratory viruses. The shrinkage of RV diversity and the near-monopoly of minor groups RV-A47 (79.5%) and RV-A49 (51.4%) suggest their fitness under pandemic conditions, in contrast to the pre-pandemic years, whereas our respiratory specimens were collected from hospitalised children at a single centre, which may be influenced by regional epidemiological characteristics. These findings present additional significance when considering parallel observations from another local hospital in Hong Kong that similarly identified a surge of RV-A47 and RV-A49 from 2020 to 2021, suggesting these genotypes had particular epidemic potential in spite of the strict social measures [17]. A similar report in the identification of RV-A47 as a predominant genotype was also found in a study from Shanghai in 2020 among children who were wearing face masks [13], but no significant shrinkage of rhinovirus diversity was observed. Moreover, the intermittent dominance pattern of RV-A49 is especially noteworthy, as this genotype previously accounted for 69.7% of the minor group RVs in 2017 and ranked among the top 10 most prevalent RV genotypes from 2015 to 2017 [18]. This historical pattern of resurgence suggests that RV-A49 possesses unique characteristics that facilitate its periodic emergence.

To understand the biological basis, we assessed the replication competence of RV-A47 and RV-A49. The direct inoculation of clinical specimens of RV-A47 and RV-A49 exhibited productive replication in HNECs, as evidenced by significant increases in viral RNA copies but not in infectious viral loads, as determined by the viral titration assay. This discrepancy implies that minor group RVs may be highly specialised to the human upper respiratory tract. However, their virus progenies generated in the HNECs replicated less efficiently in standard cell culture systems. Unlike RV-A47, which failed in H1-HeLa cells—likely due to tropism in non-respiratory cells rather than low viral loads—as RNA copy numbers were comparable to RV-A49, RV-A49 could be isolated and propagated in H1-HeLa by serial passaging in the H1-HeLa cell, and therefore yield an H1-HeLa-adapted RV-A49 for a fixed MOI comparison with the well-characterised laboratory strain RV-A16, which is widely used in RV research in vitro [19] and is one of two GMP-grade stocks approved for clinical trials [20]. Our study demonstrated that RV-A49 induces enhanced TNF/IL-1 signalling in HNECs, which suggests a hypothesis-generating link to severity, though these findings are preliminary. This study underscores the need to reassess the public health impact of minor group rhinoviruses, traditionally considered mild, as they can contribute to significant clinical consequences in paediatric populations.

Contrary to our hypothesis, RV-A49 showed reduced environmental persistence compared to RV-A16 and RV-A1B across multiple surfaces. However, as the environmental stability assay was measured using a TCID_50_ readout. This methodology may underestimate the actual environmental persistence of RV-A49, as its limited replication in H1-HeLa cells could result in lower detectable infectious titres.

The convergence of these epidemiological observations with our experimental findings, particularly the successful isolation of RV-A49 in cell culture and its distinct transcriptomic profile, strengthens the hypothesis that RV-A49 may have specific adaptations favouring its transmission under pandemic conditions and contribute to its increased detection in hospitalised children with more severe symptoms compared to community cases.

While this study provides valuable insights into the dominance and biological characteristics of minor group RVs during the SARS-CoV-2 pandemic, several limitations should be acknowledged. First, the inability to isolate RV-A47 in H1-HeLa cells prevented accurate assessment of infectious viral particle production using TCID_50_ assays. This may underestimate the transmission potential of RV-A47, as this genotype demonstrated robust viral RNA amplification in primary HNECs. This highlights a potential gap in current in vitro models for studying certain RV genotypes. Second, the clinical cohort, while informative, included only a small number of severe cases (n = 5 with tachypnoea), which limits the statistical power to draw definitive conclusions about disease severity. Nevertheless, the consistency in external surveillance [17] and mechanistic plausibility strengthen our conclusions. Third, the use of lab-adapted RV-A49 for inoculation in HNECs may have influenced the input viral genome load. H1-HeLa-adapted RV-A49 may have a lower ratio of infectious particles to viral genomes, possibly due to incomplete adaptation to H1-HeLa cells. Since MOI was calculated based on infectious titres, this could result in a higher genome load per infectious unit. Interestingly, it was found that MOI variations (1 to 0.001) do not drastically alter interferon responses in RV-infected epithelial cells [21]. Nevertheless, we are cautious about the potential impact of MOI and viral adaptation on the interpretation of host responses. Finally, real-world factors such as humidity, temperature fluctuations, and surface handling may influence viral persistence in ways not captured by our experimental setup.

## 5. Conclusions

In summary, this study identified a marked reduction in RV diversity during the SARS-CoV-2 pandemic. During the COVID-19 pandemic, RV-A47 and RV-A49 (minor groups) predominate in respiratory tract infections among children in Hong Kong, possibly due to their replication competence in primary nasal epithelial cells. The RV-A49-associated increase in severe respiratory cases in 2021 may stem from enhanced inflammatory responses.

Public health implications are clear. RV genotypes traditionally considered ‘mild’ can cause morbidity under both endemic and pandemic conditions. These findings raise important questions about how changes in human behaviour and immunity shape viral evolution and epidemiology. Future studies should explore whether minor group RVs maintain their dominance post-pandemic and investigate the molecular mechanisms underlying their selective advantage, particularly in the context of altered mucosal immunity following widespread SARS-CoV-2 infections and vaccinations.

## Figures and Tables

**Figure 1 viruses-17-01316-f001:**
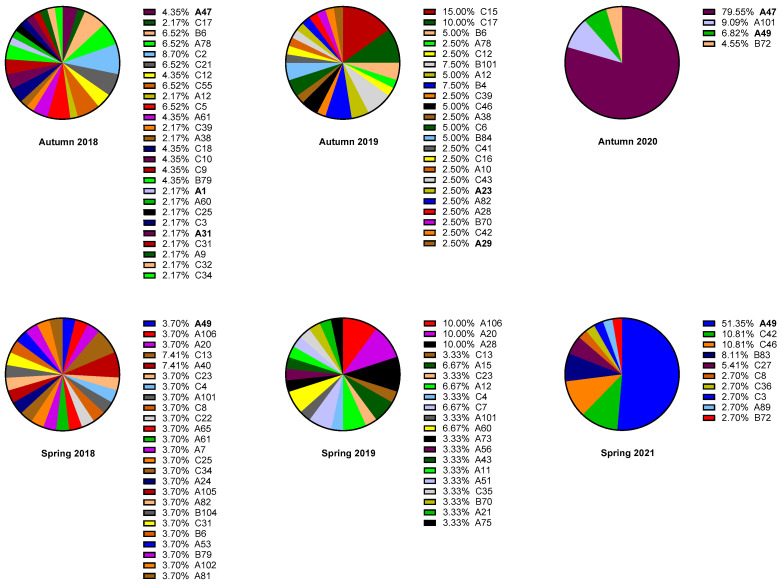
Distribution of RV genotypes during in the autumn cohort (**upper panel**) and the sprint cohort (**bottom panel**) in comparison to the monopoly of minor group during the SARS-CoV-2 pandemic. Minor group RV genotypes are denoted in bold.

**Figure 2 viruses-17-01316-f002:**
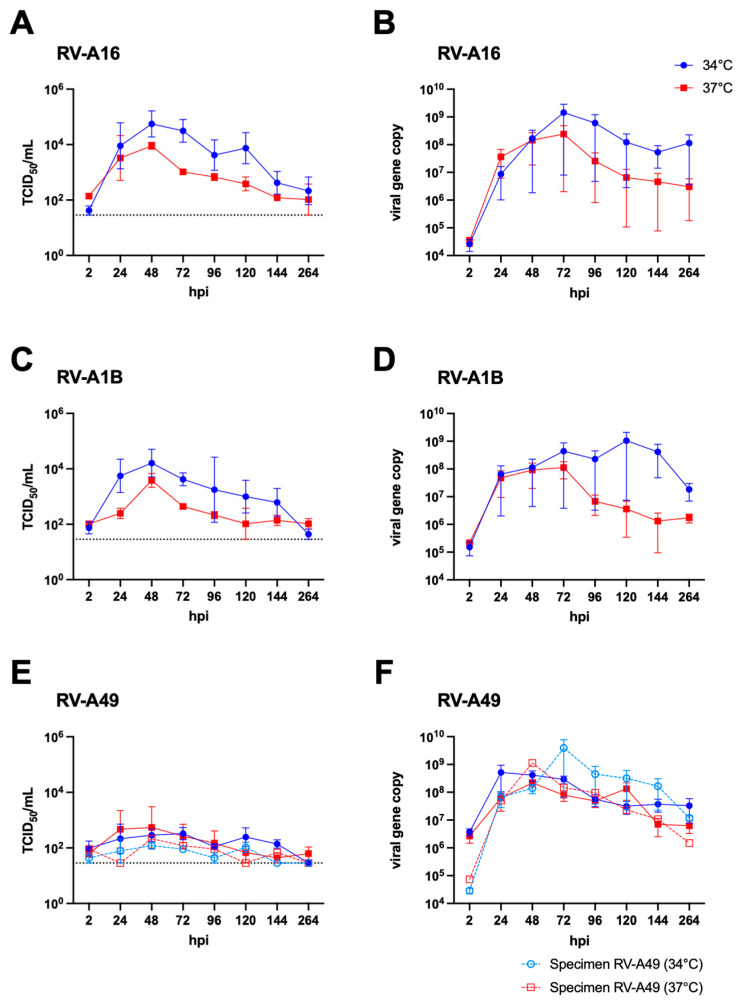
Replication kinetics of the laboratory strains RV-A16 (major) and RV-A1B (minor) and H1-HeLa-adapted RV-A49 in differentiated HNECs at 37 °C (red square) and 34 °C (blue circle) for 264 h, represented by solid lines. The infectious virus load in the supernatants of the RV-inoculated HNECs was expressed in TCID_50_/mL (**A**,**C**,**E**) and the viral gene copies (**B**,**D**,**F**) and plots show the mean and standard error of the dataset. The horizontal dotted line in the right column indicates the detection limit of the viral titration assay. For better comparison, results from direct inoculation of RV-A49 clinical specimens into the HNECs are shown using open red squares (37 °C) and open blue circles (34 °C) connected by dotted lines in (**E**,**F**).

**Figure 3 viruses-17-01316-f003:**
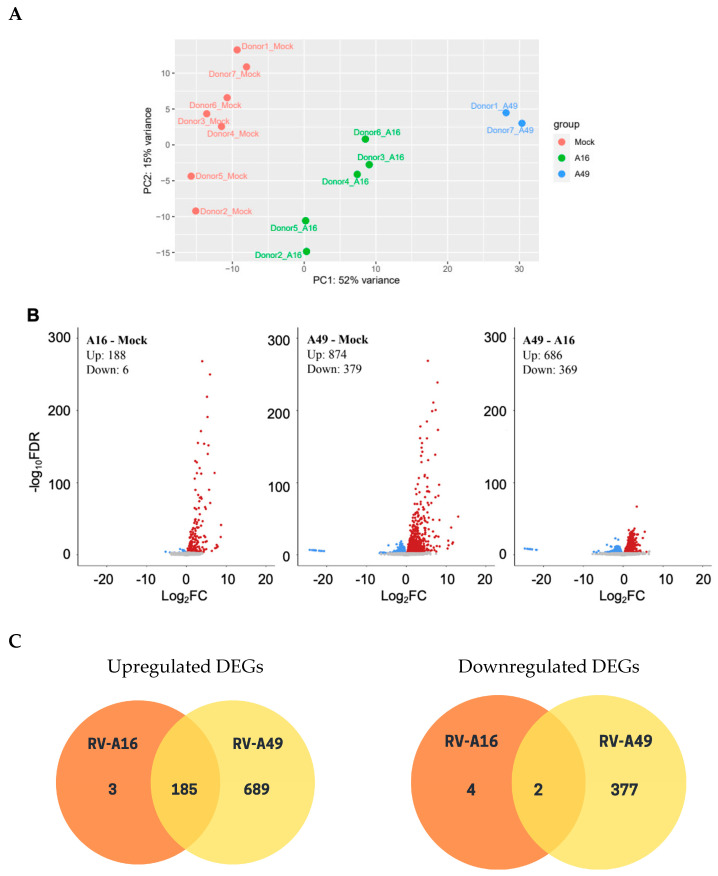
RV-A49 elicited a stronger transcriptomic response than RV-A16 in HNECs. (**A**) Principal component analysis of the bulk RNA sequencing of all samples. (**B**) Volcano plot shows differential gene expression in RV-A16- or RV-A49-infected HNECs compared to mock controls and RV-A16 compared to RV-A49-infected HNECs. (**C**) Venn diagrams show the number of upregulated and downregulated DEGs upon RV-A16 or RV-A49 infections in comparison to mock controls. (**D**) Bubble plots show the selected enriched pathways by GSEA in RV-A16 and RV-A49 infection in comparison to mock controls, with the x-axis and size of bubbles indicating the normalised enrichment score and FDR of significant pathways, respectively. (**E**) Heatmap shows the z-scores of selected genes from mock, RV-A16-, and RV-A49-infected HNECs.

**Figure 4 viruses-17-01316-f004:**
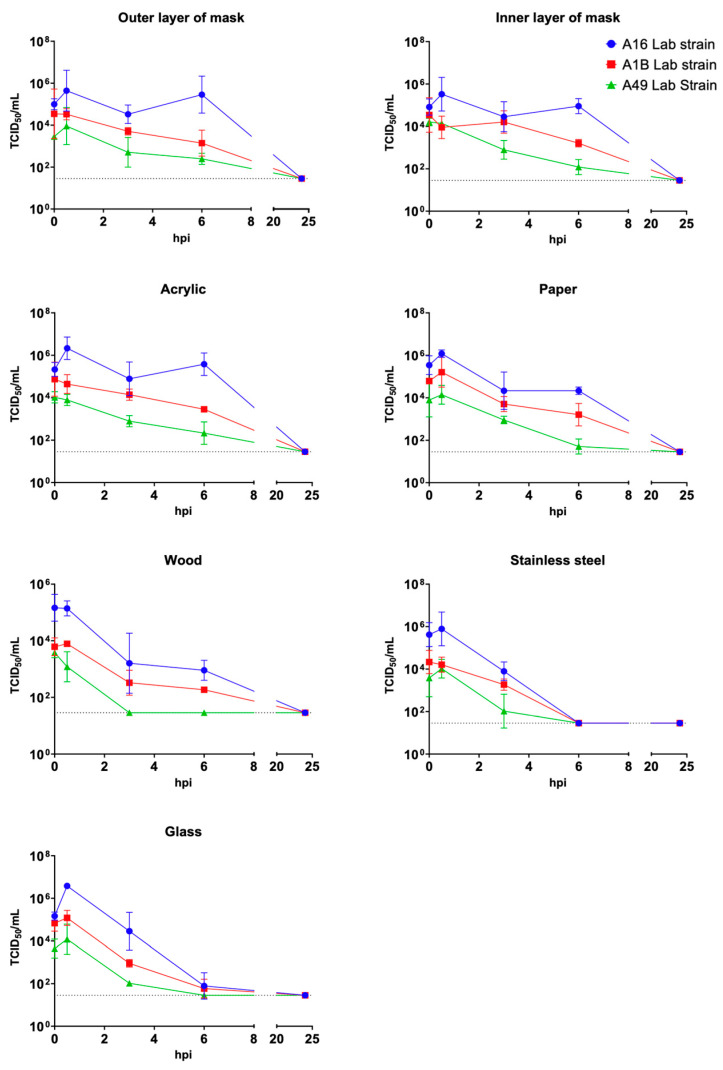
Environmental stability of RVs on various materials. Infectious titre of RVs on different surfaces over time. Data points represent the mean and SEM from duplicated viral titration assay experiments, and the dotted line indicates the detection limit of the viral titration assay.

**Table 1 viruses-17-01316-t001:** Descriptive statistics of patients in the two outbreaks of RV during SARS-CoV-2 pandemics. The first cohort started from week 44th to 48th of 2020 and the second cohort was from week 16th to week 32nd of 2021. Controls from 2018 and 2019 from the same period were selected by age and sex matching. Successful genotyped RVs per year were stratified into species of RV-A, -B, and -C and major (utilise ICAM-1 as receptor)/minor (LDLR family) grouping of RVs, with blankets indicating the percentage within the same period. Sex was analysed by Fisher’s exact test comparing the occurrence of 2020 and 2021 with pre-pandemic 2018 and 2019 baseline, and age was analysed by the Kruskal–Wallis test.

	Autumn Cohort (44th to 48th Week)				Spring Cohort (16th to 32nd Week)			
Years	2018 (n = 55)	2019 (n = 55)	2020 (n = 53)	*p*-Value	2018 (n = 34)	2019 (n = 36)	2021 (n = 41)	*p*-Value
Male (n, %)	30 (54.55)	30 (54.55)	25 (47.17)	0.677	20 (58.82)	21 (58.33)	24 (58.54)	0.999
Age (median, IQR)	3 [2, 6]	3 [2, 5]	4 [1.5, 10]	0.266	2 [1, 5]	2 [0, 4]	2 [1, 4.5]	0.805
Classification by RV-A, -B, and -C (n, %)								
RV-A	13 (28.26)	9 (22.50)	42 (95.45)	<0.0001	15 (55.56)	23 (76.67)	20 (54.05)	0.1086
RV-B	5 (10.87)	11 (27.50)	2 (4.55)	0.0198	3 (11.11)	1 (3.33)	4 (10.81)	0.7172
RV-C	28 (60.87)	20 (50.00)	0 (0)	<0.0001	9 (33.33)	6 (20.00)	13 (35.14)	0.3403
Classification by major/minor group (n, %)								
Major group	14 (30.43)	18 (45.00)	6 (13.64)	<0.0001	17 (62.96)	24 (80.00)	5 (13.51)	<0.0001
Minor group	4 (8.70)	2 (5.00)	38 (86.36)	<0.0001	1 (3.70)	0 (0)	19 (51.35)	<0.0001

## Data Availability

The raw data supporting the conclusions of this article will be made available by the authors on request, subject to ethical and institutional approval.

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
