# Peer review of "Monopoly of Minor Group Rhinovirus Infections in Hospitalised Children in Hong Kong During the SARS-CoV-2 Pandemic"

_viruses, 2025, doi:10.3390/v17101316_

Round 1
Reviewer 1 Report
Comments and Suggestions for Authors
The manuscript by Chun Sang Pun and colleagues reports intriguing findings regarding the overrepresentation of two minor receptor group rhinovirus (RV) types, RV-A47 and RV-A49, during the 2020–2021 outbreaks under stringent anti-SARS-CoV-2 public health measures.
While molecular genotyping and epidemiological data analyses support this major finding, the in vitro experiments designed to characterize the unique biological properties of A47 and A49 did not clearly demonstrate their “exceptional replication competence” in HNECs.
Major comments:
Enhanced RV replication usually positively correlates with high viral copy numbers in nasal samples and increased symptom severity. However, this correlation does not appear to hold for the dominant minor receptor group types identified in this study, which warrants further investigation.
Lines 205-210: Unsuccessful isolation of RV-A47 from 11 nasal samples in HeLa cells is not adequately explained and appears inconsistent with the manuscript’s central conclusions regarding the “monopoly of minor group RVs” and their high replication competence. RV-A47 could have been propagated in primary airway epi cells or embryonic lung fibroblasts. Were viral RNA copy numbers measured in RV-A47 NPSs before inoculation? Were they lower than in RV-A49 samples? Epidemiological data suggests that the two predominant minor group types are supposed to have similar biological properties, but this is not the case here.
Results, section 3.4. Comparison of replication kinetics of the A49 clinical isolate with HeLa-adapted lab strains A16 and A1B may not be appropriate, given that these lab strains exhibit increased replication efficiency in HeLa cells and reduced fitness to HNECs after serial passaging in HeLa-H1 cells resulting in adaptation to them. This is what the data in Figure 3 shows. Consider including replication data from clinical isolates to strengthen the comparison.
Fig. 3: The A49 input virus (viral genome copy numbers at 2h p.i.) was 1-2 log higher than that of A16 and A1B whereas the fold increase in A49 genome copy numbers was clearly lower. These results don’t support the statement that A49 has similar replication competence with A16 and A1B in vitro.
Fig.4: The observed differences in A49-induced antiviral response compared to A16 and A1B could be explained by reduced fitness of the lab strains to NHECs.
Fig. 5. These comparisons may be limited by the incomplete adaptation of the A49 isolate to HeLa cells.
Minor comments:
Line 85 and elsewhere in the text - Referring to “EV/RV” as a pathogen. While the possible reason for using this acronym is likely the nature of the diagnostic assay results with cross-reactive primers, it’s better to use “RV” instead because partial sequencing identified the pathogen as RV.
Line 116: ALI-HNECs were infected at a MOI of 0.01. Please specify the units used to calculate MOI (e.g., TCID50 or RNA copy number), as this affects interpretation of input virus levels. If TCID50 infectious units were used to calculate the virus dose, this likely explains why the A49 input virus was higher at 2h p.i.
Reviewer 2 Report
Comments and Suggestions for Authors
The article entitled “Monopoly of Minor Group Rhinovirus in Respiratory Virus Infections in Children during the SARS-CoV-2 Pandemic” by Pun et al. investigates the factors contributing to the RV outbreak in children during the SARS-CoV-2 pandemic in Hong Kong. This paper can be of broad interest to those investigating the epidemiological aspects of different rhinoviruses worldwide. I have some recommendations.
- Minor English adjustments are required.
- Line 32: Consider changing “201” by “2021.”
- In the section “Materials and Methods,” consider including subheads.
- Line 91: Consider providing the GenBank accession numbers for the study's sequences.
- In the absence of CPE, was the supernatant collected and re-inoculated?
- Consider including conditions for TCID50, such as time and volume of adsorption and number of washes with PBS following adsorption…
- Consider providing PCR cycling conditions
- If there were any symptoms other than respiratory, please report them.
- Regarding the EV/RV positive samples, what further tests were run on the samples? Did they test positive for other respiratory viruses/pathogens? Possible co-infections?
Reviewer 3 Report
Comments and Suggestions for Authors
This study investigated the epidemiological characteristics of rhinoviruses (RVs) during the COVID-19 pandemic in Hong Kong, China. The study found that despite strict public health measures, two distinct rhinovirus outbreaks occurred (RV-A47 and RV-A49 were the predominant types). RV-A49 induced a significantly stronger transcriptomic response and replicated efficiently in human nasal epithelial cells (HNECs), indicating that RV-A49 has stronger replication capacity in HNECs, which may explain its dominance during the outbreak and its impact on pediatric respiratory diseases.
This study holds certain public health significance, and below are some of my suggestions and comments:
Major:
This study should establish a better control group. For instance, it is suggested to investigate whether the transcriptomic responses and replication efficiency of the RV-A49 strain, which was identified in this study and emerged in 2020-2021, have changed compared to historical strains in HNECs, rather than merely comparing it to the standard laboratory strain RV-A16.
Minor:
1, In the introduction section, it is recommended to streamline the background description, such as compressing the details of "COVID-19 prevention and control measures", and focusing on retaining prevention and control measures related to rhinovirus transmission (such as wearing masks and hand hygiene), to avoid redundancy.
2, In Table 1, the definition of "major/minor group" needs to be explained in the table footnote.
3, It is recommended to supplement the limitations of this study, such as the fact that this study only included cases from Prince of Wales Hospital in Hong Kong, which is a "single-center study". It is necessary to state that "the results of a single-center study may be influenced by regional epidemiological characteristics". Additionally, it is important to clarify the impact of "RV-A47 being unable to be isolated", "the inability to assess the production of infectious viral particles, which may underestimate its transmission potential", etc.
4, The conclusion section suggests a more precise summary: it is recommended to revise it to read, "During the COVID-19 pandemic, RV-A47 and RV-A49 (minor group) predominate in respiratory tract infections among children in Hong Kong, and their predominance may stem from their efficient replication ability in primary nasal epithelial cells; RV-A49 is associated with an increase in severe cases in 2021 by enhancing inflammatory responses. This study suggests a need to reassess the public health impact of minor group RV".
Reviewer 4 Report
Comments and Suggestions for Authors
This manuscript examines the dominance of minor-group rhinoviruses in Hong Kong during two outbreaks under SARS-CoV-2 public health measures. The authors integrate epidemiological surveillance, viral genotyping, infection models, transcriptomic profiling, and environmental stability assays. This multidisciplinary approach might be a strength and addresses a timely and clinically relevant question. The findings have potential public health implications, but several aspects require clarification and tempering of conclusions.
1) The association between RV-A49 and greater severity in 2021 is based on only five cases with tachypnoea, four of which had RV-A49. This sample is too small for robust conclusions. Present data as preliminary. Additionally, the study enrolled individuals aged 0-17 years, rhinovirus clinical signs and symptoms may be different in newborns vs adolescents.
2) All specimens are from hospitalised children at one tertiary center. Genotype distribution may not reflect community circulation.
State this limitation explicitly and, if possible, reference additional studies.
3) Real-world conditions vary and may influence persistence. Acknowledge ecological limitations.
4) Greater TNF/IL-1 pathway activation is hypothesised to contribute to severity, but causality is unproven. Clarify that these are hypothesis-generating findings.
5) Failure to culture RV-A47 in H1-HeLa cells is interpreted as adaptation to respiratory tract; are alternative explanations (low viral load, storage degradation) possible?
Other comments:
6) Enterovirus is a genus, Rhinovirus is a species that belongs to Enterovirus genus Please revise the text accordingly.
7) Make sure to introduce all abbreviations, including SARS-CoV-2.
8) Introduction section: please mention the role and importance that rhinovirus types can have in human disease. Give a definition of RV groups.
9) Lines 50-51: references to support the effectiveness of anti-COVID-19 measures?
10) Lines 55-58 “Moreover … viruses”: move that part at the end of the Introduction, consider it as part of the aim of the study.
11) In materials and methods section there is no mention of how subjects from the pre-pandemic periods were selected.
12) Line 151: EV/RV positive or RV positive samples? It is not clear whether the authors refer to the genus or the species.
13) Data regarding the overall circulation of rhinovirus from 2020 to 2021 should be reported, in order to highlight potential increase in rhinovirus cases. What about community or outpatient data? The authors stated that several outbreaks were reported in Hong Kong but it has more than seven millions inhabitants and this study involves patients admitted at one single centre. Is it enough?
14) Lines 192-203: more data regarding the clinical outcomes should be reported in the text of the manuscript. For this aspect avoid relying too much on supplementary tables.
15) Discussion section: while quantity of references is not always a synonym for quality, the discussion section includes only four references. A proper discussion that integrates other published studies should be performed.
Round 2
Reviewer 1 Report
Comments and Suggestions for Authors
Jason Chun Sang Pun and colleagues have revised the manuscript in response to the reviewers’ comments and suggestions. The revised version is improved, but I have a couple of additional recommendations to further strengthen the manuscript and clarify the findings:
- Generalizability of Findings
The study analyzes a limited number of respiratory specimens (n=276) collected from a single hospital in Hong Kong. This raises questions about the generalizability of the findings. Further studies are needed to confirm these results and explore the mechanisms underlying the predominance of certain RV types during and after the pandemic. I recommend reflecting this limitation in the title by modifying it to:
“Monopoly of Minor Group Rhinovirus in Respiratory Virus Infections in Hospitalized Children in Hong Kong during the SARS-CoV-2 Pandemic.” - Clarification in Abstract (Lines 35–36)
The current statement regarding environmental stability may be misleading. I suggest revising it to:
“RV-A49 exhibited lower environmental stability and comparable replication efficiency to RV-A16 but induced significantly stronger transcriptomic responses, notably enhanced TNF and IL-1 signaling, in HNECs.”
Comments on the Quality of English Language
English language in revised sections.
The English in the revised sections of the manuscript should be reviewed and improved for clarity and readability.
Author Response
Comments 1: Generalizability of Findings
The study analyzes a limited number of respiratory specimens (n=276) collected from a single hospital in Hong Kong. This raises questions about the generalizability of the findings. Further studies are needed to confirm these results and explore the mechanisms underlying the predominance of certain RV types during and after the pandemic. I recommend reflecting this limitation in the title by modifying it to:
“Monopoly of Minor Group Rhinovirus in Respiratory Virus Infections in Hospitalized Children in Hong Kong during the SARS-CoV-2 Pandemic.”
Response 1: We accept the reviewer’s suggestion, and it as be amended accordingly.
Comments 2: Clarification in Abstract (Lines 35–36)
The current statement regarding environmental stability may be misleading. I suggest revising it to:
“RV-A49 exhibited lower environmental stability and comparable replication efficiency to RV-A16 but induced significantly stronger transcriptomic responses, notably enhanced TNF and IL-1 signaling, in HNECs.”
Response 2: We accept the reviewer’s suggestion, and amended the line into ‘“RV-A49 exhibited comparable replication efficiency to RV-A16 but induced significantly stronger transcriptomic responses, notably enhanced TNF and IL-1 signaling, in HNECs.” The environmental stability is controversial if we want to take into consideration of the potential underestimation of the viral load of RV-A49 in the current viral titration assay as discussed in the previous rebuttal.
To conclude, we thank the reviewer for providing us with extensive constructive comments, which led to this better version of the final manuscript.
Reviewer 2 Report
Comments and Suggestions for Authors
The majority of the issues were addressed in this version.
Author Response
We thank the reviewer for providing us with extensive constructive comments, which led to this better version of the final manuscript.
Reviewer 3 Report
Comments and Suggestions for Authors
Although the author is unable to conduct additional experiments to obtain historical isolates and make relevant comparisons, the author's explanation is acceptable. The quality of the revised manuscript has improved, and I have no further suggestions or comments.
Author Response

(The authors gave the same response as above.)

Reviewer 4 Report
Comments and Suggestions for Authors
English language editing is recommended. The text currently includes both British and American spellings.
Author Response
We thank the reviewer for providing us with this specific comment, and we checked and made the corrections in British spellings.